# Comparing the Prognostic Impacts of Delayed Administration of Appropriate Antimicrobials in Older Patients with Afebrile and Febrile Community-Onset Bacteremia

**DOI:** 10.3390/antibiotics13050465

**Published:** 2024-05-19

**Authors:** Shu-Chun Hsueh, Po-Lin Chen, Ching-Yu Ho, Ming-Yuan Hong, Ching-Chi Lee, Wen-Chien Ko

**Affiliations:** 1Department of Nursing, Meiho University, Pingtung 912009, Taiwan; x00003113@meiho.edu.tw; 2Department of Internal Medicine, National Cheng Kung University Hospital, College of Medicine, National Cheng Kung University, Tainan 70403, Taiwan; cplin@mail.ncku.edu.tw (P.-L.C.); myuan@mail2000.com.tw (M.-Y.H.); 3Department of Medicine, College of Medicine, National Cheng Kung University, Tainan 70403, Taiwan; 4Department of Adult Critical Care Medicine, Tainan Sin-Lau Hospital, Tainan 70142, Taiwan; freebrid87@gmail.com; 5Department of Nursing, National Tainan Junior College of Nursing, Tainan 700007, Taiwan; 6Department of Emergency Medicine, National Cheng Kung University Hospital, College of Medicine, National Cheng Kung University, Tainan 70403, Taiwan; 7Clinical Medicine Research Center, National Cheng Kung University Hospital, College of Medicine, National Cheng Kung University, Tainan 70403, Taiwan

**Keywords:** bacteremia, afebrile, appropriate antimicrobial therapy, outcome, afebrile bacteremia, community-onset, empirical therapy

## Abstract

Although prompt administration of an appropriate antimicrobial therapy (AAT) is crucial for reducing mortality in the general population with community-onset bacteremia, the prognostic effects of delayed AAT in older individuals with febrile and afebrile bacteremia remain unclear. A stepwise and backward logistic regression analysis was used to identify independent predictors of 30-day mortality. In a 7-year multicenter cohort study involving 3424 older patients (≥65 years) with community-onset bacteremia, febrile bacteremia accounted for 27.1% (912 patients). A crucial association of afebrile bacteremia and 30-day mortality (adjusted hazard ratio [AHR], 1.69; *p* < 0.001) was revealed using Cox regression and Kaplan–Meier curves after adjusting for the independent predictors of mortality. Moreover, each hour of delayed AAT was associated with an average increase of 0.3% (adjusted odds ratio [AOR], 1.003; *p* < 0.001) and 0.2% (AOR, 1.002; *p* < 0.001) in the 30-day crude mortality rates among patients with afebrile and febrile bacteremia, respectively, after adjusting for the independent predictors of mortality. Similarly, further analysis based on Cox regression and Kaplan–Meier curves revealed that inappropriate empirical therapy (i.e., delayed AAT administration > 24 h) had a significant prognostic impact, with AHRs of 1.83 (*p* < 0.001) and 1.76 (*p* < 0.001) in afebrile and febrile patients, respectively, after adjusting for the independent predictors of mortality. In conclusion, among older individuals with community-onset bacteremia, the dissimilarity of the prognostic impacts of delayed AAT between afebrile and febrile presentation was evident.

## 1. Introduction

Bloodstream infections (BSIs) pose a public health problem and are associated with substantial morbidity and mortality, causing an estimated 94,000 and 150,000 deaths annually in North American and Europe, respectively [1]. The incidence of these systemic infections has increased globally, irrespective of the place of acquisition [2]. Despite advances in antimicrobial treatment and critical care, BSIs remain a medical emergency with a short-time mortality rate ranging from 15 to 20% [3,4]. Notably, clinicians frequently encounter community-onset bacteremia, a common type of BSIs, in emergency departments (EDs); numerous population-based investigations reported an annual incidence ranging between 0.043% and 0.154% in the community [5], and in general, its incidence increases dramatically with age [6]. Previous investigations have indicated that appropriate antimicrobial therapy (AAT) administered promptly was crucial for reducing mortality among general populations with community-onset bacteremia [7,8,9], along with studies emphasizing that prompt AAT effectively improves the short-term prognosis in older patients [10]. 

Fever is typically a multifaceted response and generates a host defense mechanism against systemic infections. Frequently, older individuals with bloodstream infections may initially present without fever, and these afebrile episodes often manifest with atypical clinical symptoms, such as lethargy or confusion [11], which could pose a crucial challenge for first-line clinicians due to delayed diagnosis and treatment. Previous studies have demonstrated a significant association between afebrile bacteremia and delayed antimicrobial therapy or unfavorable outcomes in older patients [11,12]. However, the prognostic effect of delayed AAT in old-aged individuals with varied temperature presentations remains unclear. Accordingly, we proposed a hypothesis that there is dissimilarity in the prognostic impacts of delayed AAT administration between older patients experiencing afebrile and febrile bacteremia. 

## 2. Methods

### 2.1. Study Design

From January 2015 to December 2021, a multicenter cohort study was retrospectively conducted in three hospitals located in southern Taiwan. These hospitals were a university-affiliated medical center with 1200 beds, a teaching hospital with 460 beds, and another teaching hospital with 380 beds. This study investigated all older individuals (≥65 years) with community-onset bacteremia, including medical and surgical cases, in the emergency departments (EDs) of the study hospitals. Clinical data were reported following the guidelines outlined in the Strengthening the Reporting of Observational Studies in Epidemiology [13].

### 2.2. Patient Population

Throughout the research period, data detailing blood cultures undertaken in the EDs were collected from a computer database. The inclusion criterion was older patients (aged ≥ 65 years upon arrival at an ED) with bacterial growth on blood cultures. This study initially excluded those with contaminated blood cultures or a previous diagnosis of bacteremia before their ED visit. Additionally, patients with hospital-acquired bacteremia, uncertain death dates, or incomplete medical records were excluded from the study cohort. If a patient had multiple episodes of bacteremia, only their first episode was considered for analysis. The main outcome was to determine crude mortality within 30 days after ED arrival (i.e., bacteremia onset).

### 2.3. Data Collection

We retrospectively gathered data on patient demographics (including age, gender, and place of residence), information on any prescribed antipyretics and antibiotics prior to ED arrival, comorbidities, comorbid severity, initial infection-related syndromes, bacteremia severity (assessed by the Mortality in Emergency Department Sepsis [MEDS] score) within 24 h after ED arrival, antibiotics administered, imaging studies, surgical or radiological interventions, bacteremia sources, etiologic pathogens, and patient outcomes. To ensure accurate data collection for patients who received antipyretics and antibiotics prior to arrival at the ED, we obtained information from both chart records and post-discharge telephone contacts. Patients who could not be reached by phone or had discrepancies in the chart and telephone reports were excluded from the study and categorized as having incomplete clinical information. 

Using a predetermined record form, the data mentioned above were independently collected by two trained professionals, namely a board-certified ED physician and an infection-disease (ID) clinician. Both data collectors were unaware of the study’s aim and hypothesis. In the case of any inconsistencies in the records, the authors discussed and resolved them. 

### 2.4. Microbiological Methods

Throughout the study period, blood cultures were incubated for five days at 35 °C using the Bactec 9240 instrument from Becton Dickinson Diagnostic Systems located in Sparks, NV, USA. The etiologic pathogens were identified through matrix-assisted laser desorption ionization time-of-flight mass spectrometry and was subsequently stored in glycerol stocks at −80 °C for further susceptibility testing. This susceptibility testing was performed to determine the timing of AAT administration for each eligible patient if susceptibility to the empirically administered antimicrobials was not provided by the hospitals. The disc diffusion method for aerobes and the agar dilution method for anaerobes were used for susceptibility testing, and all susceptibility results were interpreted based on the breakpoints issued by the Clinical and Laboratory Standards Institute (CLSI) in 2023 [14]. 

### 2.5. Definitions

Bacteremia was defined as the presence of bacteria in blood cultures obtained via central or peripheral venipunctures after excluding contaminated samples. Blood cultures that contained potential contaminating microorganisms, such as Gram-positive bacilli, coagulase-negative *Staphylococcus*, *Micrococcus* species, *Propionibacterium acnes*, *Peptostreptococcus* species, and *Bacillus* species, were classified as contaminated [15]. Community-onset bacteremia referred to cases where the bacteremic episode originated in the community, including healthcare facilities not affiliated with hospitals [5]. A tympanic temperature below 38 °C (without the use of antipyretic agents) during the first 24 h after ED arrival was considered as indicative of an afebrile status, whereas a temperature measurement of 38 °C or higher was classified as febrile. Polymicrobial bacteremia was defined as the isolation of more than one microbial species from a single bacteremic episode, while monomicrobial bacteremia referred to the isolation of one or more microbial species from a single bacteremic episode.

As described earlier [8], AAT was defined based on two criteria: (i) adherence to the 2023 Sanford Guide [16] regarding the route and dosage of the administered antimicrobial agent, and (ii) in vitro activity of the antimicrobial agent against the etiologic pathogens, based on the CLSI breakpoints released in 2023 [14]. The delay in AAT was calculated as the time elapsed between the onset of bacteremia (i.e., the time of the patient’s arrival at ED triages) and the first dose of AAT administration. The administration of empirical antimicrobials that was delayed for ≥24 h was considered inappropriate. 

The origin of bacteremia was usually identified through clinical diagnosis or by pathogen isolation [17]. Primary bacteremia was characterized by cases where the source of bacteremia could not be pinpointed to a specific site. The Surviving Sepsis Campaign guideline refers complicated bacteremia to cases whose source of bacteremia is amenable to undergo source control, such as tract obstruction clearance, abscess drainage, debridement of infected necrotic tissue, device removal, and definitive source control for ongoing microbial contamination [18]. In terms of data capture, the adequacy of a specific percutaneous or surgical source control measure was jointly determined by a board-certified ED physician and an ID physician.

The definitions for comorbidities were established based on previously provided definitions [19]. Malignancies included both hematological malignancies and solid tumors. The severity of comorbidities was determined using the McCabe–Johnson classification [20]. Based on a previously described algorithm [21], the severity of bacteremia was assessed using the Mortality in Emergency Department Sepsis (MEDS) score, which incorporates measurements of nine critical components that can be obtained in an ED. Patients were categorized into mortality risk groups as follows: very low (0–4), low (5–7), moderate (8–12), high (13–15), and very high (>15), with those in the very high group (>15) considered to have critical illness. Crude mortality was used to define death from all causes.

### 2.6. Statistical Analyses

Statistical analyses were performed using the Statistical Package for the Social Sciences for Windows (Version 23.0, Chicago, IL, USA). Fisher’s exact test or Pearson’s chi-square test was used to compare categorical variables, while independent *t*-test or Mann–Whitney U test was used for continuous variables. Factors with a *p*-value of <0.05 in the univariate analysis were included in a stepwise and backward logistic regression analysis to identify independent predictors of mortality. An association of afebrile bacteremia and 30-day mortality was established using Cox regression and Kaplan–Meier survival curves after adjusting for the independent predictors of mortality.

Two methods were used to assess the prognostic effect of delayed administration of appropriate antimicrobials in patients with afebrile and febrile bacteremia. First, a multivariate regression model was used to analyze the impact of delayed AAT on mortality in patients, after adjusting for the independent predictors of mortality. Second, Cox regression and Kaplan–Meier survival curves were used to compare patients treated with inappropriate empirical antimicrobials to those treated appropriately with empirical antimicrobials, after adjusting for the independent predictors of mortality. A two-sided *p*-value of less than 0.05 was considered significant.

## 3. Results

### 3.1. Clinical Characteristics and Outcomes in Afebrile and Febrile Bacteremia

Of the 5159 older patients who had bacterial growth in their blood cultures, a cohort consisting of 3363 patients with community-onset bacteremia was captured (Figure 1). Of the 912 individuals with afebrile bacteremia, those with initial hypothermia (<36 °C) accounted for only 12.7% (116 patients). Table 1 shows the differences in bacteremia characteristics, patient demographics, length of delayed AAT, severity of illness, major sources of bacteremia, major etiologic pathogens, and mortality rates between patients with afebrile (912 patients, 27.1%) and febrile bacteremia (2451). Patients with afebrile bacteremia were more likely to be male and older; living in nursing homes; having polymicrobial bacteremia; having bacteremia caused by lower respiratory tract infections; having fatal comorbidities (McCabe–Johnson classification); having comorbidities of neurological diseases, chronic kidney diseases, or malignancies; and having etiologic pathogens of *Staphylococcus aureus*, anaerobes, or enterococci. However, a small proportion of afebrile patients had microorganisms of *Escherichia coli* and bacteremia caused by infections of the urinary tracts, biliary tracts, or bones and joints. Notably, patients with afebrile bacteremia had a more severe illness at the onset of bacteremia, a longer period of delayed AAT, and higher rates of 3-, 15-, and 30-day crude mortality than those with febrile bacteremia.

### 3.2. Prognostic Effect of Afebrile Bacteremia in Overall Patient Sample

A univariate analysis was conducted to compare clinical variables between fatal and surviving patients in terms of patient demographics, treatment for bacteremia, bacteremia severity at onset, and bacteremia characteristics (Table 2). Patients who received an inappropriate empirical antimicrobial therapy or inadequate source control were male and bedridden; had comorbidities of malignancies, neurological diseases, chronic kidney diseases, or liver cirrhosis; were critically ill (MEDS score > 15) at onset; had polymicrobial bacteremia; had bacteremia caused by lower respiratory tracts or infective endocarditis; and had etiologic pathogens of *K. pneumoniae*, *S. aureus*, anaerobes, or *Pseudomonas* species; and had a higher proportion of fatal patients. Otherwise, fatal patients exhibited lower proportions of comorbid cardiovascular disease; bacteremia caused by urinary tract infections, biliary tract infections, or liver abscess; primary bacteremia; and an etiologic pathogen of *E. coli*.

Eleven independent determinants of 30-day mortality were identified using a multivariate regression model, namely inappropriate empirical antimicrobial therapy; inadequate source control; comorbid malignancies; critical illness (MEDS score > 15) at onset; polymicrobial bacteremia; bacteremia due to lower respiratory tract infections, urinary tract infections, infective endocarditis, or liver abscess; primary bacteremia; and an etiologic pathogen of *Pseudomonas* species. After adjusting for these prognostic predictors, afebrile bacteremia was found to be associated with an increased risk of 30-day mortality (adjusted hazard ratio [AHR], 1.69; *p* < 0.001; Figure 2A) in the survival curve. In a further analysis, a decrease in temperature from 38 °C by one degree resulted in an average increase of 91% (adjusted odds ratio [AOR], 1.91; *p* < 0.001) in the 30-day mortality rate (Table 2) after adjusting for the aforementioned prognostic predictors.

### 3.3. Prognostic Impacts of Delayed AAT in Patients with Afebrile Bacteremia

A univariate analysis was conducted on 912 patients with afebrile bacteremia to compare the clinical variables between fatal and surviving patients (Table 3). The univariate analysis identified several risk factors associated with 30-day crude mortality, including bedridden status, comorbid malignancies or liver cirrhosis, critically ill status (MEDS score > 15) at onset, polymicrobial bacteremia, bacteremia originating from lower respiratory tract infections, and etiologic pathogens of *S. aureus*, anaerobes, or *Pseudomonas* species. Otherwise, the predictors against mortality included comorbid cardiovascular diseases; bacteremia caused by urinary tract, biliary tract, or skin and soft-tissue infections, primary bacteremia; and an etiologic pathogen of *E. coli*.

Using a multivariate regression model, four independent predictors of mortality were identified, namely comorbid malignancies, critical illness (MEDS score > 15) at onset, and bacteremia due to urinary or biliary tract infections. After adjusting for these four predictors, one-hour delay in AAT administration resulted in an average increase of 0.3% (AOR, 1.003; *p* < 0.001) in the 30-day crude mortality rate among patients experiencing afebrile bacteremia (Table 3). Further analysis using the survival curve showed that the inappropriate administration of empirical antimicrobial therapy had a significant prognostic impact (AHR, 1.83; *p* < 0.001), as shown in Figure 2B.

**Table 3 antibiotics-13-00465-t003:** Risk factors of 30-day crude mortality in older patients with afebrile bacteremia *.

Variables	Patient Number (%)		Univariate Analysis		Multivariate Analysis	
	Fatal, *n* = 342	Surviving, *n* = 570	OR (95% CI)	*p*-Value	Adjusted OR (95% CI)	*p*-Value
**Delayed AAT, hour**	**-**	**-**	**-**	**-**	**1.003 (1.002–1.003)**	**<0.001**
Patient demographics						
Bedridden status	128 (37.4)	156 (27.4)	1.59 (1.19–2.11)	0.001	NS	NS
Comorbidity						
Cardiovascular disease	206 (60.2)	392 (68.8)	0.69 (0.52–0.91)	0.009	NS	NS
**Malignancy**	**121 (35.4)**	**98 (17.2)**	**2.64 (1.93–3.60)**	**<0.001**	**2.24 (1.49–3.39)**	**<0.001**
Liver cirrhosis	51 (14.9)	41 (7.2)	2.26 (1.46–3.50)	<0.001	NS	NS
**Critical illness (MEDS score > 15) at onset**	**252 (73.7)**	**82 (14.4)**	**16.66 (11.91–23.31)**	**<0.001**	**12.64 (8.84–18.08)**	**<0.001**
Characteristics of bacteremia						
Polymicrobial bacteremia	63 (18.4)	61 (10.7)	1.88 (1.29–2.76)	0.001	NS	NS
Bacteremia source						
Lower respiratory tract	186 (54.4)	119 (20.9)	4.52 (3.37–6.06)	<0.001	NS	NS
**Urinary tract**	**33 (9.6)**	**169 (29.6)**	**0.25 (0.17–0.38)**	**<0.001**	**0.30 (0.19–0.49)**	**<0.001**
Skin and soft-tissue infection	27 (7.9)	69 (12.1)	0.62 (0.39–0.99)	0.045	NS	NS
**Biliary tract**	**12 (3.5)**	**78 (13.7)**	**0.23 (0.12–0.43)**	**<0.001**	**0.23 (0.11–0.49)**	**<0.001**
Primary bacteremia	5 (1.5)	22 (3.9)	0.37 (0.14–0.99)	0.04	0.35 (0.11–1.13)	0.08
Etiologic pathogen						
*Escherichia coli*	80 (23.4)	189 (33.2)	0.62 (0.45–0.84)	0.002	NS	NS
*Staphylococcus aureus*	67 (19.6)	82 (14.4)	1.45 (1.02–2.07)	0.04	NS	NS
Anaerobes	35 (10.2)	31 (5.4)	1.97 (1.20–3.28)	0.007	NS	NS
*Pseudomonas* species	20 (5.8)	18 (3.2)	1.91 (0.99–3.65)	0.049	NS	NS

AAT = appropriate antimicrobial therapy; CI = confidence interval; MEDS = Mortality in Emergency Department Sepsis; NS = not significant (after analysis using backward multivariate regression); OR = odds ratio. * Boldface indicates statistical significance with a *p*-value of ≤0.05 in the multivariate regression model.

### 3.4. Prognostic Impacts of Delayed AAT in Patients with Febrile Bacteremia

A univariate analysis was conducted on 2451 patients with febrile bacteremia to compare the clinical variables between fatal and surviving patients (Table 4). The univariate analysis identified numerous risk factors associated with 30-day crude mortality, including inadequate source control; bedridden status; comorbidities of malignancies, neurological diseases, chronic kidney diseases, or liver cirrhosis; critically ill status (MEDS score > 15) at onset, polymicrobial bacteremia; bacteremia caused by lower respiratory tract infections or infective endocarditis; and etiologic pathogens of *K. pneumoniae*, *S. aureus*, or *Pseudomonas* species. Conversely, the factors against mortality included bacteremia caused by urinary tract infections, biliary tract infections, or liver abscess; primary bacteremia; and an etiologic pathogen of *E. coli*.

Using a multivariate regression model, eight independent predictors of mortality were identified, namely inadequate source control; comorbid malignancies; critical illness at onset; polymicrobial bacteremia; and bacteremia due to lower respiratory tract infections, urinary tract infections, or infective endocarditis; and an etiologic pathogen of *Pseudomonas* species. After adjusting for these eight predictors, one-hour delay in AAT administration resulted in an average increase of 0.2% (AOR, 1.002; *p* < 0.001) in the 30-day crude mortality rate among patients with febrile bacteremia (Table 4). Further analysis using the survival curve showed that the inappropriate administration of empirical antimicrobial therapy had a significant prognostic impact (AHR, 1.76; *p* < 0.001) in febrile patients (Figure 2C).

**Table 4 antibiotics-13-00465-t004:** Risk factors of 30-day crude mortality in older patients with febrile bacteremia *.

Variables	Patient Number (%)		Univariate Analysis		Multivariate Analysis	
	Fatal, *n* = 251	Surviving, *n* = 2200	OR (95% CI)	*p*-Value	Adjusted OR (95% CI)	*p*-Value
**Delayed AAT, hour**	-	-	-	-	**1.002 (1.001–1.003)**	**<0.001**
**Inadequate source control during antimicrobial therapy**	**63 (25.1)**	**93 (4.2)**	**7.59 (5.34–10.81)**	**<0.001**	**10.70 (6.86–16.68)**	**<0.001**
Patient demographics						
Bedridden status	66 (26.2)	373 (17.0)	1.75 (1.29–2.36)	<0.001	NS	NS
Comorbidity						
**Malignancy**	**87 (34.7)**	**376 (17.1)**	**2.57 (1.94–3.41)**	**<0.001**	**2.01 (1.40–2.87)**	**<0.001**
Chronic kidney disease	74 (29.5)	496 (22.5)	1.44 (1.08–1.92)	0.01	NS	NS
Liver cirrhosis	32 (12.7)	184 (8.4)	1.60 (1.07–2.39)	0.02	NS	NS
**Critical illness (MEDS score > 15) at ED**	**149 (59.4)**	**178 (8.1)**	**16.59 (12.36–22.28)**	**<0.001**	**8.75 (6.14–12.49)**	**<0.001**
Characteristics of bacteremia						
**Polymicrobial bacteremia**	**42 (16.7)**	**166 (7.5)**	**2.46 (1.71–3.56)**	**<0.001**	**1.89 (1.17–3.04)**	**0.009**
Bacteremia source						
**Low er respiratory tract**	**113 (45.0)**	**210 (9.5)**	**7.76 (5.83–10.33)**	**<0.001**	**3.27 (2.17–4.91)**	**<0.001**
**Urinary tract**	**33 (13.1)**	**925 (42.0)**	**0.21 (0.14–0.30)**	**<0.001**	**0.38 (0.24–0.61)**	**<0.001**
Biliary tract	19 (7.6)	283 (12.9)	0.56 (0.34–0.90)	0.02	0.56 (0.30–1.04)	0.07
**Infective endocarditis**	**10 (4.0)**	**33 (1.5)**	**2.73 (1.33–5.60)**	**0.01**	**3.81 (1.54–9.44)**	**0.004**
Primary bacteremia	7 (2.8)	138 (6.3)	0.43 (0.20–0.93)	0.03	NS	NS
Etiologic pathogen						
*Klebsiella pneumoniae*	74 (29.5)	461 (21.0)	1.58 (1.18–2.11)	0.002	NS	NS
*Escherichia coli*	65 (25.9)	989 (45.0)	0.43 (0.32–0.58)	<0.001	NS	NS
*Staphylococcus aureus*	32 (12.7)	174 (7.9)	1.70 (1.14–2.54)	0.009	NS	NS
***Pseudomonas* species**	**23 (9.2)**	**61 (2.8)**	**3.54 (2.15–5.82)**	**<0.001**	**2.71 (1.36–5.42)**	**0.005**

AAT = appropriate antimicrobial therapy; CI = confidence interval; MEDS = Mortality in Emergency Department Sepsis; NS = not significant (after analysis using backward multivariate regression); OR = odds ratio. * Boldface indicates statistical significance with a *p*-value of ≤0.05 in the multivariate regression model.

## 4. Discussion

Fever is a typical response to systemic infections and has various benefits in humoral and cellular immunity, such as the mitigation of sepsis and reinforcement of the host’s defense mechanism [22]. Because fever is a common sign implying the existence of an infection, when a patient presents the febrile status, first-line clinicians routinely perform the sepsis workup and consider the possibility of infections. Nevertheless, specific populations, such as older people [11,12] and individuals with cirrhosis [23], may present with an afebrile status during episodes of bacteremia. Consistent with these previous reports [11,12,23], our cohort also indicated that this atypical presentation was associated with unfavorable prognoses. Of note, the present study highlighted that older patients experiencing afebrile bacteremia exhibited more severe adverse effects of delayed administration of appropriate antimicrobials on their short-term mortality, compared with those with febrile bacteremia. Accordingly, we reasonably believed that the poor prognosis in afebrile patients was due to the “synergetic effect” of delayed treatment and the greater impact of such a delay on prognosis. 

Due to the growing antibiotic resistance in hospitals and communities [24], as well as atypical symptoms in older patients [11], it has become an increasing challenge to effectively prescribe empirical antimicrobials for septic or infected individuals. Consistent with previous reports emphasizing the importance of prompt AAT in improving the prognoses of older individuals [10], regardless of whether they initially present with a febrile status or not, the prognostic disadvantage of delayed administration of AAT was evidenced in our cohort. More important, our study also indicated the dissimilarity in the prognostic impacts of delayed AAT administration between patients experiencing afebrile and febrile bacteremia.

Several scoring systems have been developed to predict mortality in patients with sepsis or bacteremia. However, the majorities of these predictive systems, such as the Simplified Acute Physiology Score [25], Pitt bacteremia score [8], and Acute Physiology and Chronic Health Evaluation [26], utilize the initial body temperature as an indicator of severity assessment. In our multivariate analysis, to avoid overcounting the impact of body temperature, we utilized the MEDS scores as the assessment of bacteremia and comorbid severity. Similar to a previous investigation which revealed the successful application of MEDS scores in predicting mortality risk among ED patients experiencing bloodstream infections [9], the MEDS was identified as a crucial predictor of short-term mortality in our study.

Bloodstream infections are indicative of substantial morbidity and mortality. Previous studies have shown that the annual incidence of community-onset bacteremia ranges from 43 to 154 per 100,000 individuals in the community [5], with a higher incidence among older populations [6]. The burden of community-onset bacteremia is comparable in magnitude to other medical emergencies that ED physicians manage daily, such as acute coronary syndrome, acute stroke, and major trauma. Therefore, the present study selected older patients with community-onset bacteremia as the target population.

There are several limitations to this study, primarily due to its retrospective and observational nature. First, to mitigate information bias and improve data accuracy when reviewing the medical charts, clinical data were independently collected by two well-trained physicians who were blind to the aim and hypothesis of the study, and any discrepancies in data capturing were resolved through direct discussion between the data abstractors to minimize such inconsistencies. Second, to assess the effects of delayed AAT on patient prognoses, certain patients were excluded from the analysis, such as those with uncertain fatality dates or incomplete clinical information; however, to minimize the number of excluded patients due to missing information on the period of delayed AAT, all data on etiologic pathogens were prospectively reviewed. Consequently, a trivial selection bias was expected because only a small number of patients were excluded from the analyses. Third, this study also employed a predetermined record form to comprehensively capture all clinical variables identified in previous bacteremia studies and utilized a multivariate regression model to reduce the interference of confounding factors on the study outcomes. Fourth, the adverse influence of the initially prescribed broad-spectrum antimicrobials on the development of *Clostridium difficile* infections and antimicrobial resistance was not assessed in our study. Fifth, the time of onset among patients with community-onset bacteremia was not captured in our study. However, consistent with previous ED studies that discussed the prognostic effect of empirical antimicrobial therapy [8,9], considering the time of a patient’s arrival at ED triages as the time of bacteremia onset was deemed reasonable for clinical practice. Sixth, the limitation of using a fixed temperature cut-off to define fever in our study design, along with a lack of validation using another cut-off point, should be considered. Finally, because the study hospitals were located in southern Taiwan, the findings may not be externally applicable to other communities with varying severity of comorbidities or bacteremia. Nonetheless, this study highlighted a significant difference in the prognostic effects of delayed AAT among older patients with afebrile and febrile bacteremia. Therefore, the clinical application of precision medicine or the development of a scoring system to predict bacteremic episodes among older patients with initial afebrile presentation is urgent to improve their prognosis.

## 5. Conclusions

The delayed administration of AAT significantly impacted short-term mortality in older patients with community-onset bacteremia, regardless of whether they initially presented with a febrile or afebrile status. Notably, patients with afebrile bacteremia experienced more severe adverse effects of delayed AAT on their prognoses than those with febrile bacteremia. Therefore, there is an urgent need for precision medicine to promptly detect bloodstream infections and guide the appropriateness of antibiotic administration in the aged population with initial afebrile presentation.

## Figures and Tables

**Figure 1 antibiotics-13-00465-f001:**
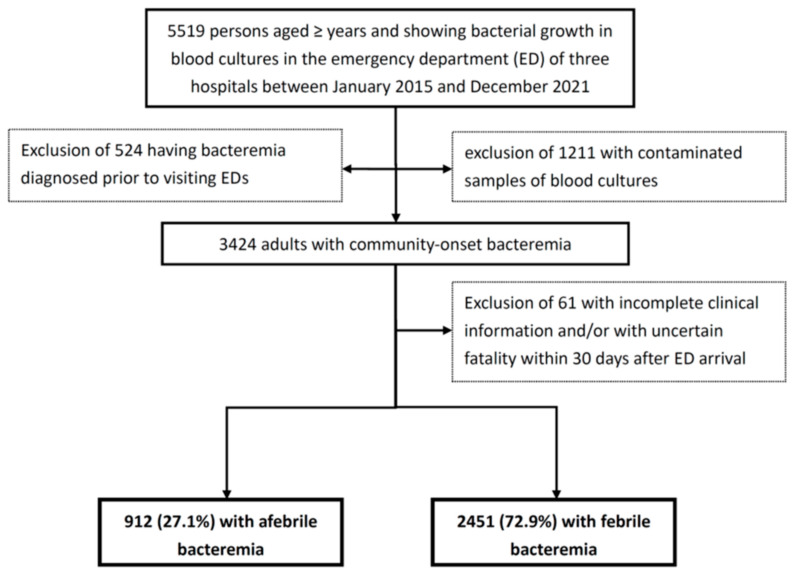
Flowchart of patient selection.

**Figure 2 antibiotics-13-00465-f002:**
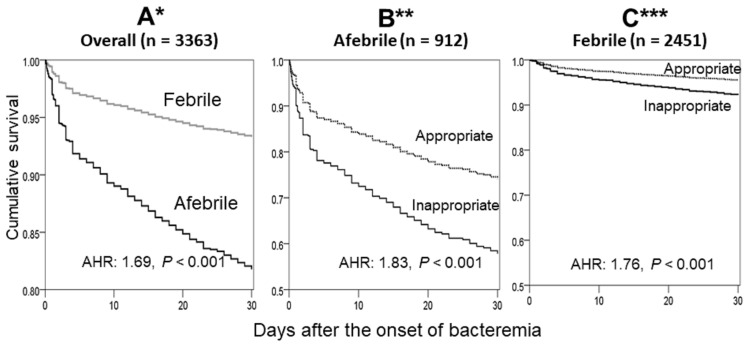
Kaplan–Meier survival curves and Cox regression comparing 30-day mortality between individuals with afebrile and febrile bacteremia in overall patient sample (**A**) * and between individuals receiving inappropriate and appropriate empirical antibiotics among those with afebrile (**B**) ** or febrile (**C**) *** bacteremia. * Adjusted for 11 independent predictors of mortality in overall patient sample (Table 2): inappropriate empirical antimicrobial therapy; inadequate source control during antimicrobial therapy; comorbid malignancies; critical illness at ED; polymicrobial bacteremia; bacteremia due to lower respiratory tract infections, urinary tract infections, liver abscess, or infective endocarditis; primary bacteremia; and an etiologic pathogen of *Pseudomonas* species. ** Adjusted for four independent predictors of mortality in patients with afebrile bacteremia (Table 3): comorbid malignancies, critical illness at ED, and bacteremia due to urinary or biliary tract infections. *** Adjusted for eight independent predictors of mortality in patients experiencing febrile bacteremia (Table 4): inadequate source control during antimicrobial therapy; comorbid malignancies; critical illness at ED; polymicrobial bacteremia; bacteremia due to lower respiratory tract infections, urinary tract infections, or infective endocarditis; and an etiologic pathogen of *Pseudomonas* species.

**Table 1 antibiotics-13-00465-t001:** Clinical manifestations and outcomes of patients with afebrile and febrile bacteremia *.

Variable	Patient Number (%)		*p*-Value
	Afebrile *n* = 912	Febrile *n* = 2451	
Patient demographics			
**Age, year, median (IQR)**	**79 (72–85)**	**77 (71–84)**	**<0.001**
**Gender, male**	**506 (55.5)**	**1178 (48.1)**	**0.001**
**Nursing-home resident**	**102 (11.2)**	**129 (5.3)**	**<0.001**
**Delayed AAT, hour, median (IQR)**	**2.7 (1.0–25.2)**	**2.0 (1.0–9.0)**	**<0.001**
**Critical illness (MEDS score > 15) at ED**	**334 (36.6)**	**327 (13.3)**	**<0.001**
Major bacteremia source			
**Low er respiratory tract**	**305 (33.4)**	**323 (13.2)**	**<0.001**
**Urinary tract**	**202 (22.1)**	**958 (39.1)**	**<0.001**
Intra-abdominal	115 (12.6)	231 (9.4)	0.07
Skin and soft tissue	96 (10.5)	228 (9.3)	0.29
**Biliary tract**	**90 (9.9)**	**302 (12.3)**	**0.049**
Bone and joint	37 (4.1)	76 (3.1)	0.17
**Primary bacteremia**	**27 (3.0)**	**145 (5.9)**	**0.001**
**Polymicrobial bacteremia**	**124 (13.6)**	**208 (8.5)**	**<0.001**
Complicated bacteremia	251 (27.5)	640 (26.1)	0.41
Major etiologic pathogen			
** * Escherichia coli* **	**269 (29.5)**	**1054 (43.0)**	**<0.001**
* Klebsiella pneumoniae*	195 (21.5)	535 (21.8)	0.83
** * Staphylococcus aureus* **	**149 (16.3)**	**206 (8.4)**	**<0.001**
streptococci	127 (13.9)	312 (12.7)	0.36
**Anaerobes**	**66 (7.2)**	**75 (3.1)**	**<0.001**
**Enterococci**	**46 (5.0)**	**86 (3.5)**	**0.04**
* Pseudomonas* species	38 (4.2)	84 (3.4)	0.31
**Fatal comorbidity (McCabe–Johnson classification)**	**279 (30.6)**	**528 (21.5)**	**<0.001**
Major comorbidity			
Cardiovascular disease	598 (65.6)	1689 (88.9)	0.07
Diabetes mellitus	390 (42.8)	1084 (44.2)	0.45
**Neurological disease**	**372 (40.8)**	**761 (31.0)**	**<0.001**
**Chronic kidney disease**	**246 (27.0)**	**570 (23.3)**	**0.03**
**Malignancy**	**219 (24.0)**	**463 (18.9)**	**<0.001**
Liver cirrhosis	92 (10.1)	216 (8.8)	0.25
Crude mortality rates			
**3-day**	**186 (20.4)**	**97 (4.0)**	**<0.001**
**15-day**	**286 (31.4)**	**193 (7.9)**	**<0.001**
**30-day**	**342 (37.5)**	**251 (10.2)**	**<0.001**

AAT = appropriate antimicrobial therapy; ED = emergency department; IQR = interquartile; MEDS = Mortality in Emergency Department Sepsis. * Boldface indicates statistical significance with a *p*-value of ≤0.05.

**Table 2 antibiotics-13-00465-t002:** Risk factors of 30-day crude mortality in older patients with community-onset bacteremia, as determined by univariate and multivariate analyses *.

Variables	Patient Number (%)		Univariate Analysis		Multivariate Analysis	
	Fatal, *n* = 593	Surviving, *n* = 2770	OR (95% CI)	*p*-Value	Adjusted OR (95% CI)	*p*-Value
**Temperature, each degree decrease from 38.0 °C**	**-**	**-**	**-**	**-**	**1.91 (1.72–2.12)**	**<0.001**
Treatment for bacteremia						
**Inappropriate empirical antimicrobial therapy ****	**159 (26.8)**	**547 (19.7)**	**1.49 (1.21–1.83)**	**<0.001**	**1.88 (1.41–2.51)**	**<0.001**
**Inadequate source control during antimicrobial therapy**	**163 (10.6)**	**94 (3.4)**	**3.38 (2.43–4.72)**	**<0.001**	**9.08 (5.93–13.92)**	**<0.001**
Patient demographics						
Gender, male	319 (53.8)	1365 (49.3)	1.20 (1.00–1.43)	0.046	1.26 (0.99–1.61)	0.06
Bedridden status	194 (32.7)	529 (19.1)	2.06 (1.69–2.51)	<0.001	NS	NS
Comorbidity						
Cardiovascular disease	366 (61.7)	1921 (69.4)	0.71 (0.59–0.86)	<0.001	NS	NS
**Malignancy**	**208 (35.1)**	**474 (17.1)**	**2.62 (2.15–3.18)**	**<0.001**	**3.43 (2.43–4.83)**	**<0.001**
Neurological disease	231 (39.0)	902 (32.6)	1.32 (1.10–1.59)	0.003	NS	NS
Chronic kidney disease	183 (27.5)	653 (23.6)	1.23 (1.01–1.50)	0.04	NS	NS
Liver cirrhosis	83 (14.0)	225 (8.1)	1.84 (1.41–2.41)	<0.001	NS	NS
**Critical illness (MEDS score > 15) at onset**	**401 (67.6)**	**260 (9.4)**	**20.16 (16.27–24.98)**	**0.001**	**9.89 (7.59–12.88)**	**<0.001**
Characteristics of bacteremia						
**Polymicrobial bacteremia**	**105 (17.7)**	**227 (8.2)**	**2.42 (1.88–3.10)**	**<0.001**	**1.51 (1.06–2.16)**	**0.02**
Bacteremia source						
**Low er respiratory tract**	**299 (50.4)**	**329 (11.9)**	**7.55 (6.19–9.20)**	**<0.001**	**1.76 (1.30–2.38)**	**<0.001**
**Urinary tract**	**66 (11.1)**	**1094 (39.5)**	**0.19 (0.15–0.25)**	**<0.001**	**0.34 (0.21–0.56)**	**<0.001**
Biliary tract	31 (5.2)	361 (13.0)	0.37 (0.25–0.54)	0.001	NS	NS
**Infective endocarditis**	**20 (3.4)**	**41 (1.5)**	**2.32 (1.35–4.00)**	**0.002**	**2.84 (1.42–5.69)**	**0.003**
**Primary bacteremia**	**12 (2.0)**	**160 (5.8)**	**0.35 (0.19–0.61)**	**<0.001**	**0.34 (0.17–0.70)**	**0.003**
**Liver abscess**	**8 (1.3)**	**105 (3.8)**	**0.35 (0.17–0.72)**	**0.003**	**0.22 (0.09–0.52)**	**0.001**
Etiologic pathogen						
*Klebsiella pneumoniae*	150 (25.3)	581 (21.0)	1.28 (1.04–1.57)	0.02	1.33 (0.98–1.80)	0.06
*Escherichia coli*	145 (24.5)	1178 (42.5)	0.44 0.36–0.54)	<0.001	NS	NS
*Staphylococcus aureus*	99 (16.7)	256 (9.2)	1.97 (1.53–2.53)	<0.001	NS	NS
Anaerobes	47 (7.9)	94 (3.4)	2.45 (1.71–3.52)	<0.001	NS	NS
***Pseudomonas* species**	**43 (7.3)**	**79 (2.9)**	**2.66 (1.82–3.90)**	**<0.001**	**2.02 (1.16–3.50)**	**0.01**

CI = confidence interval; NS = not significant (after analysis using backward multivariate regression); MEDS = Mortality in Emergency Department Sepsis; OR = odds ratio. * Boldface indicates statistical significance with a *p*-value of ≤0.05 in the multivariate regression model. ** delayed AAT ≥ 24 h.

## Data Availability

The data presented in this study are available on request from the corresponding author.

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
