# Peer review of "Comparing the Prognostic Impacts of Delayed Administration of Appropriate Antimicrobials in Older Patients with Afebrile and Febrile Community-Onset Bacteremia"

_antibiotics, 2024, doi:10.3390/antibiotics13050465_

Round 1
Reviewer 1 Report
Comments and Suggestions for Authors
Title – would add community-onset to description of bacteremia in title.
Introduction – occasionally the word choice or sentence structure reads awkwardly to an English first-language reader, e.g. page 2, second paragraph, second sentence.
Methods:
Study design – was this a retrospectively identified cohort or a prospective cohort with only the data collected retrospectively. This could be more clearly stated in 2.1 of methods. The study included patients with community-onset bacteremia. Since the study is focused on timing of antibiotics from time of onset of illness, it would be important here to define “onset of bacteremia” as this is an arbitrary timepoint. It is unclear if the authors decided to define the onset of bacteremia as the date of admission. Reference the STROBE tool used for clinical data reporting.
Patient population. Define older individuals here – was your population only individuals >/= to 65 years of age or some other cut off age used (it appears to be based on Figure 1)? In this paragraph, it states that “Of the older patients having bacterial growth on blood cultures, the study initially excluded…”. Does this mean of the patients of older age or the patients at the beginning of the study period? What was done subsequently that was different?
Data collection. No comments.
Definitions. This is also where time of onset of bacteremia could be included since this is where delay in therapy is defined. A delay of > 24 hours is considered inappropriate – assuming this means 24 hours from presentation to the emergency department not from time of onset of presenting symptoms. This could be more explicitly described.
Results.
Not a criticism of results but a choice of wording suggestion.
More commonly use lower respiratory tract infection versus low respiratory to describe deep lung infections.
Authors have chosen to refer to outcome groups as fatal and survival. This changes in Table 2 to Deaths versus Survival. I would try to be consistent. In the subsections 3.3 and 3.4 it might be clearer to the reader to compare the risk factors associated with death to those associated with survival rather than focusing on saying which risk factors had a lower association with death; for those identify them as having better survival prognosis.
Discussion.
Limitations – no discussion of possible impact of surrogate definition of an inappropriate delay in time to AAT given that the authors are defining this as 24 h after presentation to ED, or the limitation of defining onset of bacteremia as also time of presentation to ED. Additionally, no discussion of the possible limitation of using a fixed temperature to define fever.
Would like to see in the discussion any comment if relevant related to bacteremia due to Pseudomonas aeruginosa infection outcomes due to delayed time to AAT. Was bacterial resistance a factor that impacted initial appropriate empiric therapy significantly such that further investigation into choice of initial empiric antimicrobials is warranted?
Would like to see a comment in general on 60% of afebrile patients with bacteremia dying compared to 11.4% of febrile patients with bacteremia. Was this within expected parameters of authors at onset?
Figure 1. Flow chart – good
Table 1.
Instead of defining critical illness at ED in footnotes put MEDS > 15 into the table
Figure 2 – KM curves; good
Table 2
Critical illness at onset – change to MEDS score > 15
Table 3; just use MEDS score > 15
Comments on the Quality of English LanguagePlease see general comments.
Author Response
Please refer to the letter to reviewer 1.

Reviewer 2 Report
Comments and Suggestions for Authors
This is a thorough review of the subject population, i.e. elderly patients presenting to ED with community onset bacteremia (BSI). However, the conclusion is not very insightful. How would precision medicine impact the findings of this retrospective study. What would be more helpful to better define the patients with BSI who are afebrile such that the reader has some sort of takeaway message, based on your data, that would better identify the bacteremic patient and identify the most appropriate empirical antibiotic therapy. Better data are presented in the figures showing Kaplan Meier survival curves. The discussion should focus on separating temperature deviation from delay in appropriate antibiotic selection.
Other things which could make this study better include:
1. breakdown of community onset BSI using conventional definitions which contrast health care related vs non health care related infections. This would also help the reader discern those at risk for BSI due to the health care related status.
I was confused by your apparently different definitions for appropriate antibiotic therapy, sometimes defined by empirical use and other by appropriate use based on pathogen identification and susceptibility results. A conventional singles analysis of time to effective antibiotics would be preferred.
3. description of the patient population should include palliative or supportive care vs aggressive treatment. It is implied that lack of source control was due to the patients being too frail for surgery. Were these patients included even if antibiotic therapy was not considered appropriate or withdrawn during the 30-day mortality window?
4. how many of the afebrile patients were hypothermic? This is a key vial sign for considering sepsis.
5. minor edit, the term "low" respiratory tract was used in table and text. Should be "lower" respiratory tract.
My reading of the abstract was underwhelming, and I believe could be strengthened to fit your conclusion, that afebrile BSI patients have a worse outcome (survival). Your initial comparison in the abstract is for hours of delated appropriate therapy, and although statistically significant the difference in the Odds Ratios between afebrile and febrile (1.003 vs 1.002 is hardly clinically meaningful. Similarly, the OR for mortality of 1.83 vs 1.76 is not clinically helpful. In your text, after the regression analysis you provide the adjusted HR for 30 day survival in panel A for the overall population: 1.69, p<0.001. I suggest that this AHR belongs in the abstract.
Comments on the Quality of English Language
Minor editing of English language required
Author Response
Please refer to the letter of response-to-reviewer 2
